# Fecal Microbiota Analysis in Cats with Intestinal Dysbiosis of Varying Severity

**DOI:** 10.3390/pathogens11020234

**Published:** 2022-02-10

**Authors:** Nikolay Bugrov, Pavel Rudenko, Vladimir Lutsay, Regina Gurina, Andrey Zharov, Nadiya Khairova, Maria Molchanova, Elena Krotova, Marina Shopinskaya, Marina Bolshakova, Irina Popova

**Affiliations:** 1Department of Veterinary Medicine, Peoples’ Friendship University of Russia (RUDN University), 117198 Moscow, Russia; 1042180166@rudn.ru (N.B.); rudenko-pa@rudn.ru (P.R.); recaro21@bk.ru (V.L.); krotova-ea@rudn.ru (E.K.); shopinskaya-mi@rudn.ru (M.S.); bolshakova-mv@rudn.ru (M.B.); 2Biological Testing Laboratory, Branch of Shemyakin-Ovchinnikov Institute of Bioorganic Chemistry of the Russian Academy of Sciences (BIBCh RAS), 142290 Pushchino, Russia; 3Department of Technosphere Safety, Peoples’ Friendship University of Russia (RUDN University), 117198 Moscow, Russia; gurina-rr@rudn.ru (R.G.); zharov-an@rudn.ru (A.Z.); khairova-ni@rudn.ru (N.K.); 4Department of Foreign Languages, Peoples’ Friendship University of Russia (RUDN University), 117198 Moscow, Russia; molchanova-ma@rudn.ru

**Keywords:** microbiota, biotope, dysbiosis, intestines, irrational antibiotic therapy, diagnostics, cats

## Abstract

Recent studies have shown that the gut microbiota plays an important role in the pathogenesis of gastrointestinal diseases in various animal species. There are only limited data on the microbiome in cats with varying grades of dysbiosis. The purpose of the study was a detailed analysis of the quantitative and qualitative fecal microbiota spectrum in cats with intestinal dysbiosis of varying severity. The data obtained indicate that, depending on the dysbiosis severity in cats, the intestinal microbiome landscape changes significantly. It has been established that, depending on the dysbiosis severity, there is a shift in the balance between the Gram-positive and Gram-negative bacterial pools and in the nature of the isolation of specific bacteria forms, in the amount of obligate microbiota isolation, as well as individual facultative strains. When analyzing the serotyping of *E. coli* cultures isolated at various grades of intestinal dysbiosis severity, differences were found both in the isolation amount of various serotypes from one animal and in the prevalence of certain serotypes for each disease severity. A retrospective analysis of the fecal microbiota sensitivity in cats with dysbiosis to antibacterial drugs showed that, depending on the disease severity, the number of isolates sensitive to antibiotics increases significantly.

## 1. Introduction

The optimization of veterinary services, the reduction of morbidity and mortality of animals, and the effective prevention and improvement of measures to combat various diseases play a decisive role in improving the quality of life of animals [1,2,3,4,5,6,7]. Recently, in cities, there has been a significant increase in the number of dogs and cats, which often become full-fledged family members. The value of these pets has also increased, not so much in the economic as in the bioethical aspects. A number of undeniable advantages are inherent in small animals: they improve our mood, reduce emotional and physical stress, help their owners cope with depression, create a feeling of exaltation, as well as a warm and cozy atmosphere in the hearth. In addition, dogs and cats contribute to the psychological development of children with autism, cerebral palsy, and developmental delays and improve the quality of life for lonely, elderly people and patients with cardiovascular pathologies [8,9,10,11]. However, despite the rapid development of veterinary medicine, many issues of diseases in small domestic animals, which are caused by associations of opportunistic and saprophytic microbiota, as well as the improvement of diagnostic approaches, prevention, and therapy, remain poorly understood [12,13,14,15,16].

The relationship between microbiota and their host does not take place in isolation with each individual but associatively as part of the microbiocenosis, which is an open, unstable, constantly changing group of evolutionarily ecologically related microorganisms [15,17,18,19]. Under the influence of environmental factors, unsatisfactory nutrition, systematic violations of elementary veterinary and sanitary rules of keeping, failure to fully implement measures to prevent infectious diseases, and irrational antibiotic therapy, the body’s immune system may suffer, and as a result, various immunodeficiency states arise in the animal [11,13,20]. As a result, not only conditionally pathogenic microflora are activated but even saprophytic microorganisms. This situation creates ample opportunities for various combinations of bacterial formation in the biotopes of the animal organism, leading to poor quality microbiocenoses [21,22,23].

The study of the intestinal microbiota’s role has convincingly shown that they are the most important component of the protective intestinal barrier, which controls the interaction of the macroorganism and its environment [11,24,25,26]. The gastrointestinal microbiome is a multicomponent consortium of bacteria, archaea, fungi, protozoa, and viruses that inhabit the intestines of all mammals. Studies have shown that the gut microbiome is involved in a number of physiological processes vital to host health, including energy homeostasis, metabolism, gut epithelial health, immunological activity, neurobiological development, and psychoemotional status [27,28,29,30,31,32]. In recent years, interest in the interaction of the gut microbiota with the host has increased due to many conclusions about the influence of gut bacteria on the health and development of various diseases and pathological conditions. As a rule, the occurrence of complications is associated with the pathological translocation of intestinal bacteria or endotoxins, which is the result of intestinal barrier dysfunction [33,34,35,36,37,38,39].

New approaches to bacterial identification in veterinary medicine have shown that the gastrointestinal tract microbiota in dogs and cats, like humans, are a very complex ecosystem that controls immune responses and the development of allergic and inflammatory diseases [22,40,41,42,43,44,45]. However, at the moment, in the available literature, there is no information on diagnostic approaches for intestinal dysbiosis of varying severity in cats. Improving the intestinal dysbiosis in cats, as per course, in our opinion, contributes to the improvement of predicting the outcome of pathology, as well as methods of its correction and prevention. In connection with the above, the purpose of this study is a detailed analysis of the quantitative and qualitative spectrum of fecal microbiota in cats with intestinal dysbiosis of varying severity.

## 2. Results

During the initial examination of the experimental animals, special attention was focused on the severity of clinical signs in gastrointestinal tract disorders. Thus, in cats, at the initial visit with intestinal dysbacteriosis syndrome, most often, an unpleasant odor from the oral cavity was recorded (89.1%), as well as loss of appetite and decreased intestinal motility (56.5% each), dry skin and mucous membranes (54.3%), and constipation (45.6%).

When comparing the clinical signs of dysbacteriosis according to the severity, it was found that for all 15 cats with a severe stage of dysbiotic disorders of the intestine, dry skin and mucous membranes, decreased appetite, and an unpleasant odor from the oral cavity were characteristic. In addition, five cats (33.3%) had pruritus, four (26.7%) had diarrhea, and 11 (73.3%) had diarrhea alternating with constipation. It should be noted that in the group of cats with grade 3 severity of dysbacteriosis, the pathology of the gastrointestinal tract manifested itself in the form of severe depression, and liquid or unformed feces were noted, while the frequency of defecation was 5–10 times a day. Furthermore, an increase in signs of dehydration and intoxication syndrome was noted, which are characterized by a forced lying position of animals and hyporexia or anorexia. These signs made it possible to determine the severe (decompensated) course of intestinal dysbacteriosis.

In 16 of 46 sick cats (34.8%), subcompensated intestinal dysbacteriosis was recorded, the clinical manifestations of which were an unpleasant odor from the oral cavity (93.7%), anorexia, and dry skin and mucous membranes (50.0%). When analyzing the nature of the stool, it was found that eight animals (62.5%) had constipation, five (31.3%) had liquid feces, and three animals (6.3%) had alternating constipation and diarrhea. In cats with immature soft stools, the frequency of defecation was 3–4 times per day. Signs of body dehydration for this stage of dysbacteriosis were insignificant. Clinical methods in cats in most cases revealed slight weakness, and hyporexia was also noted at a normal body temperature. According to the indicated clinical picture of the disease, the presence of moderate severity of the course of intestinal dysbacteriosis was ascertained.

In 15 of 46 sick cats (32.6%), the clinical picture of the disease was characterized by clear consciousness, normothermia, normorexia, and the animals actively and voluntarily changing their position and moving freely in space, and vomiting and signs of dehydration were not registered. At the same time, in 11 individuals (73.3%), an unpleasant odor from the oral cavity was observed; in three (20.0%), there a decrease in appetite; and in two (13.3%), dry skin and mucous membranes were observed. When analyzing the nature of the stool in cats with a mild degree of dysbacteriosis, it was found that 13 animals (86.7%) had constipation, and two noted the presence of formed feces with an uneven color. In this case, a mild or compensated course of intestinal dysbacteriosis of grade 1 was diagnosed. Etiological verification, depending on the grade of intestinal dysbiosis in cats, showed that with compensated grade 1 of intestinal dysbiosis severity in experimental animals, in 100.0% of cases, the etiological factor was an alimentary factor. In the case of the second severity level, the most often recorded were drug-induced (7 (43.7%)) and postoperative (4 (25.0%); less often were invasive (3 (18.8%)) and alimentary (2 (12.5%)) dysbiosis. On the contrary, when a severe (decompensated) grade of intestinal dysbiosis appeared in animals, the etiological factor in most cases of the total (14 (93.3%)) revealed the manifestation of drug dysbiosis. In only one animal with a severe grade of intestinal dysbiosis, the etiological factor was invasion.

When conducting microbiological studies of fecal samples taken from 46 cats with a preliminary intestinal dysbiosis diagnosis, 304 bacterial strains of 23 species were isolated. The results of microbiological studies are shown in Table 1.

The presented data indicate that enteric bacterium, staphylococci, streptococci, pseudomonas, citrobacteri, and enterobacteria, as well as fungi of the genus *Candida*, were most often isolated from fecal samples in cats with intestinal dysbiosis. It was found that, depending on the dysbiosis severity in cats, the microbial landscape of intestinal eubiosis changed significantly. Thus, with grade 3 dysbiosis in fecal samples, there was a decrease in isolates of staphylococci (with the exception of *S. aureus*), streptococci (with the exception of *S. uberis*), lactobacilli (with the exception of *L. acidophilus*), and bifidobacteria, when compared with dysbacteriosis of the 1st severity. It should be noted that only *B. adolescentis* strains were isolated from representatives of the genus *Bifidobacterium* in grade 3 intestinal dysbiosis in three (4.3%) of the total number of isolates. The indicated decrease in isolates in case of grade 3 dysbacteriosis occurred against the background of an increase in the isolation of *P. aeruginosa*, *P. vulgaris*, *C. freundii*, and *C. albicans* fungi. It must be said that with intestinal dysbiosis of both grades 1–3 of severity, the number of isolates of enteric bacterium, enterobacteria, and bacilli was practically at the same level.

According to the results of a biological test on white mice, it was found that all 361 strains of microorganisms that we isolated during the research did not have pathogenic properties and did not cause the death of laboratory animals.

The results of serological identification of isolated *E. coli* cultures with intestinal dysbiosis of varying severity in cats are presented in Table 2. It was found that in cats with intestinal dysbiosis, the serological pool of *E. coli* was quite diverse and had significant differences from serotypes isolated from fecal samples from clinically healthy animals. Thus, nine serotypes of enteric bacterium were isolated from six healthy cats. At the same time, *E. coli* serogroups O1, O4, and O9 were isolated more often, in three (33.4%), two (22.2%), and two (22.2%) cultures, respectively. In cases of intestinal dysbiosis of grade 1, 25 serogroups of E. coli were isolated from 15 cats. At the same time, O1, O2, O9, O83, and O116 were isolated more often, with three (12.0%) each of the total number of isolates. In cases of dysbacteriosis of grade 2, 30 serogroups of enteric bacterium were isolated from 16 animals; more often, O18 and 022 were isolated, with five (16.7) cultures each, and of O8, O26, and O101, there were four (13.3%) cultures each, of the total number of enteric bacterium.

With intestinal dysbiosis of grade 3 in 15 cats, we identified 39 serogroups of enteric bacterium. At the same time, O18 was most often isolated, with 7 (17.9%), followed by O8 and O26, with 6 (15.4%), and O22 and O101, with 5 (12.8%) serogroups.

The frequency dynamics of *E. coli* serogroups isolation in cats, depending on the intestinal dysbiosis severity, found its imprint in Figure 1.

The data presented in Figure 1 indicate that in clinically healthy cats, one serogroup was isolated from fecal samples in three cases (50.0%), and in three animals (50.0%), two serological groups of Escherichia coli were isolated. Depending on the intestinal dysbiosis severity in cats, the number of isolated Escherichia serogroups from fecal samples also changed. Thus, with grade 1 dysbiosis, two *E. coli* serogroups were isolated in six (40.0%) animals, and one *E. coli* serogroup was isolated in four (26.7%) cats. In the case of grade 2 dysbacteriosis, two *E. coli* serogroups were isolated in seven (43.8%) animals, of the total number of animals in the group. With intestinal dysbiosis of grade 3, three *E. coli* serogroups were isolated most often in nine (60.0%) cats, out of the total number of pets in the group. It should be noted that only in animals with grade 3 dysbiosis, in one cat (6.7%), isolation from fecal samples of four serogroups of enteric bacterium was recorded.

At the same time, significant quantitative changes in the intestinal microbiota in diseased cats were verified by bacteriological methods, which are shown in Figure 2 and Table 3.

The data shown in Figure 2 indicate that with intestinal dysbiosis in cats, there was a significant decrease in the number of representatives of the genus *Lactobacillus*. It should also be noted that, depending on the dysbiosis severity, there was a sharper decrease in their number. Thus, with intestinal dysbiosis of the grades 1–3, a significant decrease in the number of lactobacilli was recorded of 1.21, 1.61, and 2.86 times, respectively, when compared with their amount in fecal samples from clinically healthy animals. Similar results were obtained when analyzing the concentration of bifidoflora representatives in feces samples from patients with dysbacteriosis. It was found that with grade 1 dysbiosis, there was a significant (*p* < 0.05) decrease in the number of bifidobacteria by 1.18 times, from 9.36 ± 0.42 to 7.90 ± 0.33 lg CFU/cm^3^. In the case of intestinal dysbiosis of grade 2, a significant (*p* < 0.001) decrease in the number of bifidobacteria by 2.66 times was observed in experimental animals, from 9.36 ± 0.42 to 3.52 ± 0.64 lg CFU/cm^3^. In cases of grade 3 dysbiotic intestinal disorders in cats, a highly significant (*p* < 0.001) decrease in the number of genus *Bifidobacterium* representatives by 4.50 times, from 9.36 ± 0.42 to 2.08 ± 0.51 lg CFU/cm^3^, was recorded when compared with animals of the control group.

In addition, a significant increase in the titer of the genera *Streptococcus* spp. (*p* < 0.01), *Escherichia* spp. (*p* < 0.01), *Citrobacter* spp. (*p* < 0.05), *Klebsiella* spp. (*p* < 0.01), *Proteus* spp. (*p* < 0.05), and *Pseudomonas* spp. (*p* < 0.01) microorganisms and the genus *Candida* fungi (*p* < 0.05) by 2.74, 1.33, 4.26, 5.08, 2.90, 4.52, and 3.39 times, respectively, was found when compared with similar indicators in the control group.

The sensitivity of microbiota isolated from fecal samples of cats with intestinal dysbiosis to antibacterial drugs is shown in Table 4, Table 5 and Table 6.

The data presented in the tables indicate that with different grades of dysbiotic disorders, the sensitivity of the microbiota to antibacterial agents varies significantly.

It has been established that in cats with grades 2 and 3 intestinal dysbiosis, there is a decrease of bacterial cultures sensitive to antibacterial agents. Thus, in cats with dysbiotic disorders from the second and third experimental groups, a decrease in sensitivity to benzylpenicillin by 5.9% and 12.5%, to methicillin by 2.2% and 6.8%, to amoxicillin by 7.5% and 16.2%, to cefazolin by 4.6% and 8.2%, to ceftriaxone by 5.0% and 12.1%, to gentamicin by 7.6% and 12.3%, to lincomycin by 3.7% and 11.9%, and to enrofloxacin by 2.9% and 13.8%, respectively, were recorded. The sensitivity of the genus *Candida* fungi isolated during dysbiosis in cats to antimycotic drugs is shown in Figure 3.

The data shown in the figure indicate that the isolated fungal strains exhibited high sensitivity to fluconazole (100.0% of cultures were sensitive). All fungi were susceptible to amphoterricin B, except for one strain (10.0%) isolated from a cat with grade 3 intestinal dysbiosis. The isolates showed a lower sensitivity to the antimycotic intraconazole. All isolates were susceptible to it, except for two (28.6%) strains isolated from animals from grade 2 intestinal dysbiosis and two (20.0%) fungi from cats with grade 3 intestinal dysbiosis.

## 3. Discussion

New facets of the complexity and multicomponents of the intestinal microbiota, as well as their shifts, leading to disorders of the gastrointestinal tract, became available for us [8,24,45]. A clinician’s blind therapeutic approach, only with correctly diagnosed intestinal dysbiosis in cats, in practice, does not always lead to a speedy recovery. Thus, in some animals, the use of antimicrobial agents leads to a speedy recovery, and in some patients with gastrointestinal diseases, the appointment of antibiotics aggravates intestinal dysbiosis and leads to the development of some complications.

A number of recent studies have revealed the features of the intestinal microbiome in cats in normal conditions, as well as in various pathological conditions [11,14,18,33]. However, these studies did not analyze the fecal gut microbiota in cats with dysbiosis of varying severity. Therefore, a detailed analysis of the quantitative and qualitative spectrum of fecal microbiota in cats with intestinal dysbiosis is an urgent direction for research work.

In the current study, we compared the fecal microbiome between healthy cats and cats with varying grades of gut dysbiosis. The data obtained indicated that, depending on the dysbiosis severity, the microbial landscape of intestinal eubiosis changed significantly. Thus, with grade 3 dysbiosis in fecal samples, there was a significant decrease in isolates of staphylococci (except *S. aureus*), streptococci (except *S. uberis*), lactobacilli (except *L. acidophilus*), and bifidobacteria, when compared with grade 1 dysbacteriosis. This decrease occurred against the background of a significant increase in the isolation of *P. aeruginosa*, *P. vulgaris*, *C. freundii*, and *C. albicans* fungi. Analyzing the differences in the microbial landscape in intestinal dysbiosis of varying severity, it should also be noted that with the most severe grade 3, we did not identify representatives of *S. saprophyticus*, *S. epidermidis*, *L. plantarum*, *L. xylosus*, *B. animalis,* and *B. bifidum* cultures. A similar trend in changes in the species spectrum of fecal microbiota in intestinal dysbiosis in cats was observed by other researchers [41,44]. However, the authors did not recommend differentiating intestinal dysbiosis in cats according to severity.

Interesting data were the analysis of the qualitative ratio of fecal microbiota in cats with varying severity dysbiosis, which is shown in Figure 4.

The data presented in the figure indicate that, depending on the intestinal dysbiosis severity, the ratio of Gram-positive and Gram-negative microflora in fecal samples taken from experimental cats differed significantly. Thus, in clinically healthy cats and animals with grade 1 dysbiosis, the Gram-positive microbiota prevailed over the Gram-negative one, with 40 (63.5%)/23 (36.5%) and 61 (51.3%)/58 (48.7%), respectively. On the contrary, in cases of grades 2 and 3 dysbacteriosis, Gram-negative microorganisms significantly prevailed in the intestinal microbiota, with 41 (33.3%)/82 (66.7%) and 27 (19.8%)/109 (80.2%), respectively. Thus, with the most severe decompensated grade of intestinal dysbiosis in cats, Gram-negative microbiota were significantly higher than normal values by 43.7%. Similar results were obtained by the authors [43], indicating the predominance of Gram-negative fecal microbiota in cats with inflammatory bowel disease in alimentary small cell lymphoma.

It was found that in the case of intestinal dysbiosis in cats, the serological pool of enteric bacterium is quite diverse and has significant differences with serotypes isolated from clinically healthy animals’ fecal samples. In addition, for each intestinal dysbiosis severity in cats, there is a serological spectrum of E. coli strains, which may need to be taken into account in diagnostic approaches. Discussing the data obtained, it should be noted that at grade 2 and 3 intestinal dysbiosis in cats, we did not isolate O1, O2, O4, O9, O113, and O116 from fecal samples, and in the most severe grade 3 dysbacteriosis, we did not isolate O83 and O114 serogroups. It should also be said that, depending on the intestinal dysbiosis severity in cats, the amount of discharge of *E. coli* serogroups from fecal samples also changed significantly. Thus, with grade 1 dysbiosis, two serogroups of *E. coli* were isolated more often. With grade 3 intestinal dysbiosis in nine cats, three serogroups of *E. coli* were isolated, and in one cat, isolation from fecal samples of four serogroups of *E. coli* was recorded. We obtained these data for the first time.

We also found that in cats with intestinal dysbiosis, there was a significant decrease in the number of the genera *Lactobacillus* and *Bifidobacterium* in fecal samples, which, depending on the dysbiosis severity, was aggravated and gained maximum values, with the most severely decompensated being grade 3 (*p* < 0.001). This occurred against the background of a significant increase in the titer of the genera *Streptococcus* spp. (*p* < 0.01), *Escherichia* spp. (*p* < 0.01), *Citrobacter* spp. (*p* < 0.05), *Klebsiella* spp. (*p* < 0.01), *Proteus* spp. (*p* < 0.05), and *Pseudomonas* spp. (*p* < 0.01) microorganisms and the genus *Candida* (*p* < 0.05) fungi, when compared with similar indicators of cats in the control group. It should also be noted that in the control group, representatives of the genera *Proteus* and *Pseudomonas* were not isolated from fecal samples (Table 3). A number of researchers obtained similar results [11,18,22,37,40].

The main purpose of assessing the bacterial sensitivity to antibiotics is to predict their effectiveness. The use of unified methods for determining the sensitivity and approaches to interpreting the results is a prerequisite for the formation of a unified analysis and data exchange system [11]. Therefore, for a more detailed analysis of any differences depending on the intestinal dysbiosis severity in cats, we also conducted such studies. The studies showed that with various grades of intestinal dysbiotic disorders in cats, the sensitivity of the microbiota to antibacterial agents varied significantly. It was found that with the most severe (decompensated) grade of intestinal dysbiosis in cats, there was a significant decrease in the amount of intestinal microbiota sensitive to antibacterial agents. We are sure that this interesting trend needs to be further studied using the serial dilution method and more accurate statistical data processing methods. It should be noted that all bacteria isolated from intestinal dysbiosis in cats with grades 1, 2 and 3 were susceptible to the antibiotic of the cephalosporin series IV generation cefepime and the antimicrobial agent from the group of fluorinated quinolones, gatifloxacin. Earlier, during a clinical examination of sick animals, it was found that when the most severe (decompensated) grade of intestinal dysbiosis occurred, the etiological factor in most cases (14 (93.3%)) revealed a drug-induced factor, namely irrational use of antibiotics. Perhaps this fact is the reason for the differences in antibiotic sensitivity of isolates at different grades of dysbiosis severity in cats. We obtained these data for the first time.

It should be noted that the sensitivity of isolated fungi to antimycotics at different grades of different grades of dysbiosis severity did not differ and was stable.

Thus, a detailed microbiological approach to classifying the intestinal dysbiosis severity in cats is a very important step in predicting the pathology course, as well as in further developing an algorithm for an individual therapeutic approach for each patient. The data obtained should be taken into account by practicing veterinary specialists when developing an effective strategy for combating intestinal dysbiosis in domestic cats. Further research is needed to evaluate other articulations of the gut microbiome, such as parasites, viruses, and protozoa, in order to better understand the microbial dynamics in dysbiosis, depending on the severity.

## 4. Materials and Methods

### 4.1. Animal Subjects and Study Design

The studies were carried out during 2019–2021 on the basis of private clinics of veterinary medicine: “Avetura”, “Epiona” (Moscow), and “V mire s zhivotnymi” (Serpukhov). Examination of the cats and selection of the biomaterial for research was carried out in accordance with the International Bioethical Standards, the provisions of the IV European Convention “On the Protection of Vertebrate Animals Used for Experimental and Other Scientific Purposes” (ETS 123, 1986), as well as legislative documents of the Russian Federation on conducting experiments on animals.

Cats were selected for the experiment as they entered the veterinary clinic for the first time. The study included 46 adult animals, aged from 2 to 6 years, of mixed sex with intestinal dysbiosis syndrome. All cats were kept in apartments in an urban setting. The control consisted of clinically healthy individuals (n = 6), aged 2 to 6 years, of mixed sex, which were examined with the written consent of their owners before routine vaccination. The cats were fed a commercial dry, balanced adult animal feed, Purina Pro Plan, three times a day.

The diagnosis was made in a complex way, taking into account the data of anamnesis, clinical examination, and microbiological studies. The intestinal dysbacteriosis severity (grade 1: compensated; grade 2: subcompensated; grade 3: decompensated) was assessed on the basis of clinical and laboratory studies.

Intestinal dysbacteriosis of grade 1 (compensated) in cats: the level of consciousness was within the normal range; in most cases it was accompanied by constipation; often, the appearance of an unpleasant odor from the oral cavity; and in rare cases, a decrease in appetite and dryness of the external integument, no signs of dehydration, and a body temperature within normal limits.

Intestinal dysbacteriosis of grade 2 (subcompensated) in cats: the level of consciousness was slightly depressed; there was an unpleasant odor from the oral cavity; dry skin and mucous membranes were often manifested, as well as hyporexia. This was accompanied by constipation and diarrhea and, in rare cases, alternating constipation and diarrhea. There was dehydration within 5%, with possibly a slight increase in body temperature. 

Intestinal dysbacteriosis of grade 3 (decompensated) in cats: the level of consciousness was depressed; necessarily, the presence of anorexia, an unpleasant odor from the oral cavity, dry skin and mucous membranes, and itching were also possible. In most cases, it was accompanied by alternating constipation and diarrhea and, in some cases, diarrhea. Dehydration was within 10%, with possibly an increase in body temperature or hypothermia.

Depending on the intestinal dysbacteriosis severity, the animals were divided into the following groups: group 1, cats with grade 1 dysbacteriosis (n = 15); group 2, animals with grade 2 intestinal dysbacteriosis (n = 16); group 3, pets with grade 3 intestinal dysbiosis (n = 15).

The inclusion criteria was clinically healthy animals, as well as cats with intestinal dysbiosis of varying severity.

The exclusion criteria was poor compliance of cat owners with the recommendations of doctors on therapy and feeding of animals.

When collecting an anamnesis from the owners, special attention was paid to the etiological factors for the occurrence of intestinal dysbiosis in their pets. In the absence of an understanding of the cause of the dysbacteriosis syndrome, additional special laboratory studies were performed. A stool analysis for eggs and segments of helminths and simple cysts was performed to exclude or confirm invasive dysbiosis. Prior to initiating the study, microscopic examination of feces after centrifugation based on sugar solution was performed on the feces of all cats to evaluate for parasite eggs, cysts, and oocysts. In the laboratory diagnosis of feces, macroscopic and microscopic methods and modern instrumental methods were used. Parasep (Apacor LTD, London, United Kingdom) disposable concentrators are designed for detecting the concentration of intestinal parasites by centrifugation through a specialized filter (a modification of the formalin ether method). Additionally, methods of flotation (floating) were used, which are based on the difference in the specific weight of the flotation solution and the helminth eggs; the specific weight of the flotation solution is higher, and, as a result, the helminth eggs float to the surface in liquids and are found in the surface film. A commercially available direct fluorescent antibody assay was used to evaluate for the presence of *Giardia* spp. cysts and *Cryptosporidium* spp. oocysts. To confirm or exclude drug-induced dysbiosis, the determination of the antibiotic in the blood by the Kozmin–Sokolov method was used. For this, two rows of test tubes were placed in a rack. In one of them, a dilution of a reference antibiotic was prepared and, in the other, an experimental liquid. Then, a suspension of test bacteria prepared in Giss’s medium with glucose was added to each tube. The concentration of the antibiotic was determined by multiplying the highest dilution of the test liquid which inhibited the growth of test bacteria by the minimum concentration of the reference antibiotic which inhibited the growth of the same test bacteria. In addition to the above, panleukopenia in cats was necessarily excluded, which is also accompanied by symptoms similar to dysbiosis. A parvo test kit was used for the rapid diagnosis of parvovirus infection in carnivores by detecting the antigen of the causative agent of feline panleukopenia in the feces of infected animals.

### 4.2. Microbiological Research

When carrying out microbiological studies from the selected material isolated from cats with a Pasteur pipette, inoculations were made on nutrient media. For yeast-like fungi, Sabouraud’s glucose agar was used; for staphylococci, a peptone salt medium, yolk-salt agar, and MPA were used; for enterobacteria, Endo agar, Ploskirev’s medium, and bismuth sulfite agar were used; for bifidobacterial, Maurocyllmed medium and lactic acid were used. The inoculations were again incubated in a thermostat at 37–38 °C for 24 h, and in the absence of growth, the dishes were kept for up to 3 days.

After studying the cultural and morphological properties of all individual typical colonies, subcultures were made in the same test tubes and incubated at 37–38 °C for 24 h. The obtained pure cultures of bacteria were checked for mobility in preparations of a crushed drop using phase contrast microscopy in a darkened field of view and subjected to identification. For quantitative bacteriological studies of the fecal samples, they were performed on 1.0 g of the studied substances in further studies. From the first test tube, which was considered a 10−1 dilution, further tenfold dilutions were prepared up to 10−10. Then, from each tube, 0.1 cm^3^ of the resulting mixture was inoculated into Petri dishes on the surface of solid nutrient media (Endo, MPA, Sabouraud, Ressel, Blaurocca nutrient medium, MRS, bismuth sulfite agar, yolk-salt agar, and solid nutrient medium “PSL”). The semi-liquid medium based on sodium thioglycolate (HiMedia, Maharashtra, India) created anaerobic conditions. The crops were incubated in a thermostat at 37 °C for 7 days. The presence of microbial growth in nutrient media was assessed visually by the appearance of turbidity, film, sediment, and other changes. At the end of the incubation period, fixed preparations were prepared from test tubes with visible growth of microorganisms in color according to the Gram and Leffler method, followed by microscopy and identification. The number of microorganisms in 1.0 cm^3^ of the starting material (C) was calculated by the following formula and expressed in logarithms with base 10:C = (N/V) × K
where N is the average number of colonies in 1 bacteriological dish; V is the volume of the suspension, which is applied during inoculation on the surface of the agar; and K is the multiplicity of dilution.

The morphology of bacteria was studied in smears stained according to Gram and Romanovsky–Giemsa staining. Further identification of biochemical properties was carried out in accordance with the “Bergey’s Identifier for Bacteria”. Gram-negative rods that gave a positive result in the test for the presence of catalase and a negative result in the test for cytochrome oxidase, oxidized and fermented glucose (in Hugh–Leifson’s medium), and reduced nitrates were assigned to the *Enterobacteriaceae* family. All isolated cultures were inoculated on Giss media with glucose, maltose, lactose, mannose, sucrose, mannitol, and dulcite. Gram-positive rod-shaped bacteria were additionally subcultured onto His medium with galactose, salicin, fructose, and arabinose. To determine the catalase activity of microorganisms, the bacterial mass removed with a loop from the agar surface was suspended in a drop of 3% hydrogen peroxide on a slide. For further identification, the genus and species representatives of the *Enterobacteriaceae* family in the culture were subcultured onto Olkenitsky’s medium in a long variegated row, which included media with mannitol, maltose, sucrose, xylose, rhamnose, dulcite, sorbitol, salicin, Rochelle salt (d-tartrate), milk with litmus, and beef-extract broth for the study of indole, as well as tests for the utilization of citrate and acetate, the formation of H2S with methyl-mouth, and the presence of phenylalanine deaminase. In Gram-negative rod-shaped bacteria, the fermentation of carbohydrates such as inositol and sorbitol was additionally determined using paper indicator systems (Nizhniy Novgorod, Russia); the utilization of sodium citrate and malonate; the production of hydrogen sulfide, indole, and acetylmethylcarbinol; and the presence of ornithine decarboxylase, lysine decarboxylase, phenylalanine deaminase, and β-galactosidase enzymes. To eliminate motility in cultures of the *Proteus* genus, 96° ethanol was poured into bacteriological dishes with MPA before the studies, kept for 3–5 min, and then the ethanol was removed. Determination of *E. coli* serogroups was carried out using a set of “O-coliaglutinating serum” (“Armavir Biofabrika”, Russia). To identify bacteria of the *Pseudomonadaceae* (Pseudomonas) family, the culture was subcultured on King B medium in a test tube beef-extract broth and grown in a thermostat at a temperature of 42° C. For the differentiation of bacteria of the *Staphylococcus* genus from the *Streptococcus* genus, the presence of catalase was determined. The differentiation of the *Staphylococcus* genus from the *Micrococcus* genus used a glucose oxidation–fermentation test (Hugh–Leifson’s medium). To identify the species of bacteria of the *Staphylococcus* genus, tests were carried out for the presence of coagulase; the oxidation of mannitol, galactose, maltose, lactose, and sucrose; and the ability to grow in the presence of 10% NaCl. To identify the bacteria species of the *Streptococcus* genus, tests were carried out for the ability to grow in air at 10 °C and 45 °C at pH 9.6 in the presence of 6.5% NaCl and 40% bile, for hemolysis, and for sugar fermentation. To determine the pathogenicity of isolated cultures, three white mice weighing 14–16 g were injected intraperitoneally with 1 billion microbial cells for each strain of the microorganism. Laboratory animals were observed for 5 days. Cultures were considered pathogenic if one or more mice died within five days of infection. At the death of the animal, a bacteriological study of the pathogenic material selected from the parenchymal organs was performed to compare the isolate with the introduced pathogen.

### 4.3. Antibiotic Sensitivity

Determination of the bacterial sensitivity to antibacterial drugs and antimycotics was carried out using the disk diffusion method according to the European Committee on Antimicrobial Susceptibility Testing recommendations (EUCAST; V 8.0, 2020). A total of 10 antibiotics (benzylpenicillin, methicillin, amoxicillin, cefazolin, ceftriaxone, cefepime, gentamicin, lincomycin, enrofloxacin, and gatifloxacin) and 3 antimycotics (amphoterricin B, fluconazole, and intraconazole) were used as test drugs. When evaluating the results, sensitive isolates were considered with a growth retardation of more than 18 mm; low sensitive were with growth retardation of 11–18 mm, and resistant had a growth retardation of less than 10 mm.

### 4.4. Statistical Analysis

The obtained research results were processed statistically and presented in the form of tables and figures. All calculations were performed using the STATISTICA 7.0 statistical program. The arithmetic mean (Mean), root mean square error (SE), and standard deviation (SD) were calculated. The reliability of the difference in indicators between the indicators of the control and experimental groups was calculated using the Mann–Whitney method (*—*p* < 0.05; **—*p* < 0.01; ***—*p* < 0.001).

## 5. Conclusions

In conclusion, the results of this study revealed significant differences in the fecal microbiota spectrum in cats with varying grades of gut dysbiosis. It was found that with dysbacteriosis in the intestinal microbiome in cats, they were significant, and with decompensated grade 3 of severity, there were profound qualitative and quantitative changes. It has been established that, depending on the dysbiosis severity, there is a shift in the balance between the Gram-positive and Gram-negative bacterial pools and in the nature of the isolation of specific bacteria forms, in the amount of obligate microbiota isolation, as well as in individual facultative strains. When analyzing the serotyping of E. coli cultures isolated at various grades of intestinal dysbiosis severity in cats, differences were revealed both in the number of various serotypes from one animal and in the prevalence of certain serotypes for each pathologic behavior. A retrospective analysis of the fecal microbiota sensitivity in cats with dysbiosis to antibacterial drugs showed that, depending on the severity, the number of isolates sensitive to antibiotics increased significantly.

## Figures and Tables

**Figure 1 pathogens-11-00234-f001:**
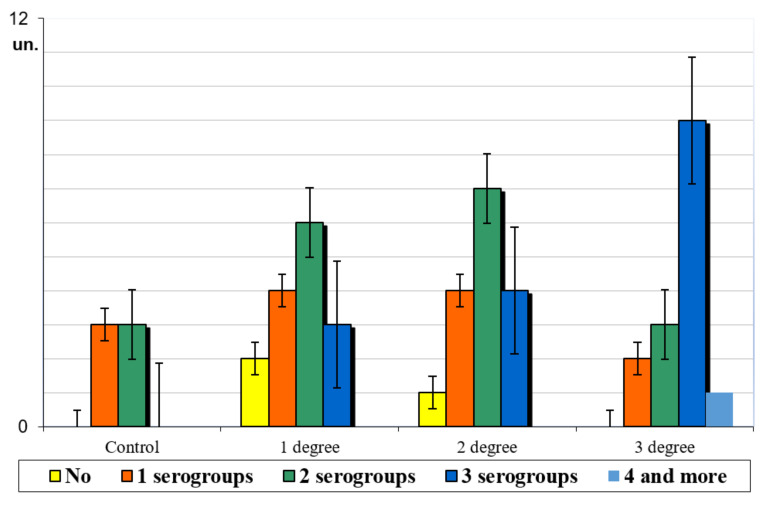
Frequency dynamics of *E. coli* serogroups isolation in cats depending on the intestinal dysbiosis severity.

**Figure 2 pathogens-11-00234-f002:**
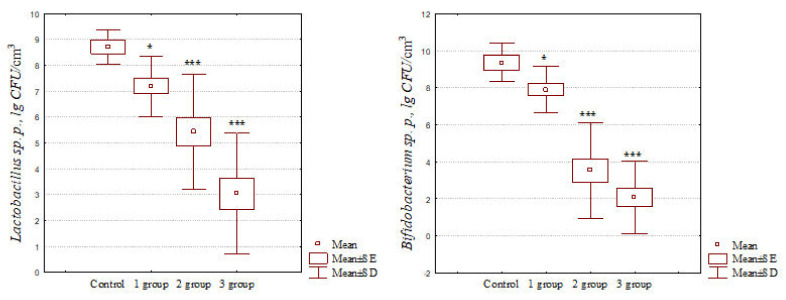
Lactobacilli and bifidobacteria concentrations (lg CFU/cm^3^) in feline feces, depending on the dysbiosis severity. Mean is the arithmetic mean; SE is the standard error; SD is the standard deviation; *, *** is the reliability of the difference between the experimental and control groups (Mann–Whitney test, *—*p* < 0.05; ***—*p* < 0.001).

**Figure 3 pathogens-11-00234-f003:**
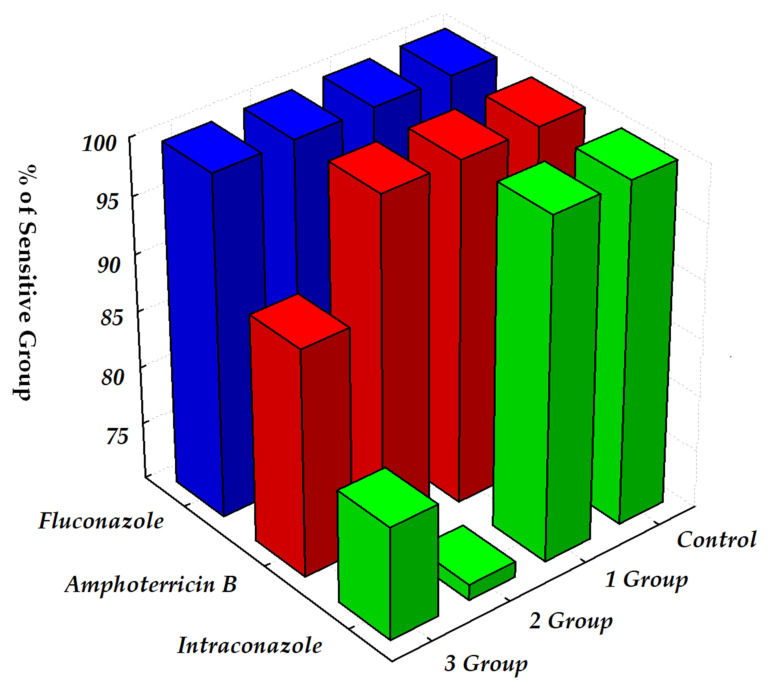
Sensitivity of the isolated genus *Candida* fungi (n = 21) to antimycotics.

**Figure 4 pathogens-11-00234-f004:**
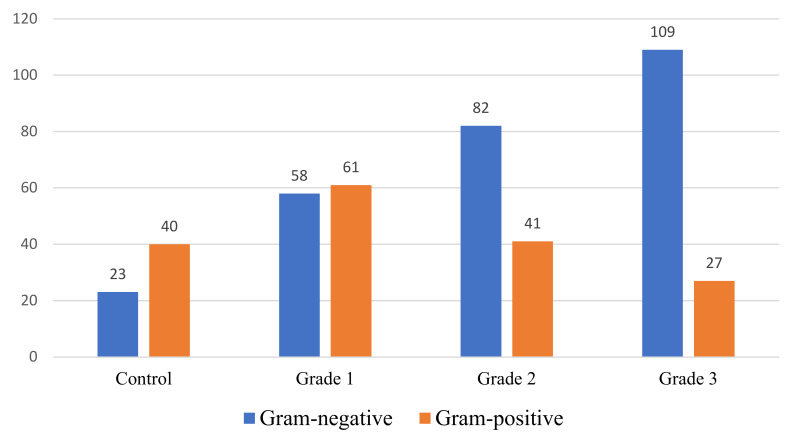
The ratio of fecal microbiota in cats with varying dysbiosis severity.

**Table 1 pathogens-11-00234-t001:** Fecal microbiota species spectrum in intestinal dysbiosis in cats (n = 46).

Type of Microorganism	Isolates from Fecal Samples								Total	
	Healthy Cats (n = 6)		With Intestinal Dysbiosis							
			Grade 1 (n = 15)		Grade 2 (n = 16)		Grade 3 (n = 15)			
	Abs	%	Abs	%	Abs	%	Abs	%	Abs	%
*Staphylococcus saprophyticus*	4	7.0	4	4.1	1	1.0	−	−	9	2.5
*Streptococcus intermedius*	2	3.5	7	7.2	1	1.0	1	0.9	11	3.1
*Staphylococcus epidermidis*	1	1.7	−	−	2	2.0	−	−	3	0.8
*Staphylococcus aureus*	−	−	6	6.2	5	5.0	7	6.5	18	4.9
*Streptococcus agalactiae*	3	5.3	4	4.1	3	3.0	1	0.9	11	3.1
*Streptococcus faecalis*	4	7.0	4	4.1	3	3.0	1	0.9	12	3.3
*Streptococcus uberis*	1	1.7	1	1.0	5	5.0	8	7.5	15	4.1
*Escherichia coli*	6	10.5	13	13.5	15	15.0	15	14.1	49	13.7
*Pseudomonas aeruginosa*	−	−	3	3.1	5	5.0	12	11.3	20	5.5
*Proteus vulgaris*	−	−	1	1.0	2	2.0	6	5.6	9	2.5
*Citrobacter freundii*	2	3.5	4	4.1	8	8.0	11	10.3	25	6.9
*Enterobacter aerogenes*	3	5.3	6	6.2	11	11.0	10	9.3	30	8.3
*Klebsiella oxytoca*	2	3.5	3	3.1	6	6.0	5	4.7	16	4.4
*Klebsiella pneumoniae*	−	−	−	−	−	−	2	1.9	2	0.6
*Bacillus subtilis*	2	3.5	4	4.1	5	5.0	9	8.4	20	5.5
*Lactobacillus plantarum*	3	5.3	5	5.1	3	3.0	−	−	11	3.1
*Lactobacillus rhamnosus*	3	5.3	5	5.1	2	2.0	1	0.9	11	3.1
*Lactobacillus xylosus*	2	3.5	3	3.1	3	3.0	−	−	8	2.2
*Lactobacillus acidophilus*	3	5.3	4	4.1	5	5.0	5	4.7	17	4.7
*Bifidobacterium adolescentis*	3	5.3	5	5.1	4	4.0	3	2.8	15	4.1
*Bifidobacterium animalis*	5	8.8	4	4.1	2	2.0	−	−	11	3.1
*Bifidobacterium bifidum*	6	10.5	9	9.4	2	2.0	−	−	17	4.7
*Candida albicans*	2	3.5	2	2.2	7	7.0	10	9.3	21	5.8
Total	57	100	97	100	100	100	107	100	361	100

“−” means negative result.

**Table 2 pathogens-11-00234-t002:** Serological identification of isolated *E. coli* cultures in cats with dysbiosis.

Serogroup	Isolates from Fecal Samples								Total	
	Healthy Cats (n = 6)		With Intestinal Dysbiosis							
			Grade 1 (n = 15)		Grade 2 (n = 16)		Grade 3 (n = 15)			
	Abs	%	Abs	%	Abs	%	Abs	%	Abs	%
O1	3	33.4	3	12.0	−	−	−	−	6	5.8
O2	1	11.1	3	12.0	−	−	−	−	4	3.9
O4	2	22.2	2	8.0	−	−	−	−	4	3.9
O8	−	−	−	−	4	13.3	6	15.4	10	9.7
O9	2	22.2	3	12.0	−	−	−	−	5	4.8
O18	−	−	1	4.0	5	16.7	7	17.9	13	12.7
O22	−	−	−	−	5	16.7	5	12.8	10	9.7
O26	−	−	−	−	4	13.3	6	15.4	10	9.7
O83	−	−	3	12.0	2	6.8	−	−	5	4.8
O101	−	−	2	8.0	4	13.3	5	12.8	11	10.8
O111	−	−	−	−	1	3.3	3	7.7	4	3.9
O113	1	11.1	2	8.0	−	−	−	−	3	2.9
O114	−	−	2	8.0	1	3.3	−	−	3	2.9
O116	−	−	3	12.0	−	−	−	−	3	2.9
O119	−	−	1	4.0	3	10.0	3	7.7	7	6.8
O127	−	−	−	−	1	3.3	4	10.3	5	4.8
Total	9	100	25	100	30	100	39	100	103	100

“−” means negative result.

**Table 3 pathogens-11-00234-t003:** Fecal microbiota concentration (lg CFU/cm^3^) in cats with dysbiosis, depending on the disease severity.

Genus	Control Group (n = 6)	Animals with Intestinal Dysbiosis		
		Grade 1 (n = 15)	Grade 2 (n = 16)	Grade 3 (n = 15)
*Staphylococcus* spp.	3.23 ± 0.88	4.87 ± 0.32	3.77 ± 0.87	4.98 ± 0.97
*Streptococcus* spp.	2.56 ± 0.85	4.53 ± 0.46 *	4.62 ± 0.74	7.01 ± 0.59 **
*Escherichia* spp.	6.19 ± 0.41	5.76 ± 0.46	6.72 ± 0.48	8.21 ± 0.26 **
*Bacillus* spp.	1.32 ± 0.86	1.74 ± 0.58	1.71 ± 0.68	3.09 ± 0.93
*Enterobacter* spp.	2.19 ± 1.04	3.23 ± 0.69	3.73 ± 0.69	4.73 ± 0.93
*Citrobacter* spp.	1.25 ± 0.85	1.58 ± 0.51	2.59 ± 0.69	5.33 ± 0.91 *
*Klebsiella* spp.	1.26 ± 0.84	2.08 ± 0.79	1.97 ± 0.68	6.41 ± 0.76 **
*Proteus* spp.	0	0.49 ± 0.25	0.71 ± 0.35	2.90 ± 0.80 *
*Pseudomonas* spp.	0	0.27 ± 0.17	1.08 ± 0.53	4.52 ± 0.75 **
*Candida* spp.	1.35 ± 0.86	2.18 ± 0.61	2.24 ± 0.71	4.58 ± 0.90 *

*—*p* < 0.05; **—*p* < 0.01 (significance of the difference between the experimental and control groups; Mann-Whitney test, *—*p* < 0.05; **—*p* < 0.01).

**Table 4 pathogens-11-00234-t004:** Isolated microbiota sensitivity (n = 97) to antibacterial drugs in the case of grade 1 dysbiosis in cats.

Antibacterial Drugs	Antibiotic Susceptibility of Isolated Microbiota					
	Sensitive		Low Sensitive		Resistant	
	Abs	%	Abs	%	Abs	%
Benzylpenicillin	61	62.9	24	24.7	12	12.4
Methicillin	71	73.2	22	22.7	4	4.1
Amoxicillin	82	84.5	12	12.4	3	3.1
Cefazolin	86	88.6	11	11.4	−	
Ceftriaxone	97	100.0	−	−	−	−
Cefepim	97	100.0	−	−	−	−
Gentamicin	49	50.6	16	16.5	32	32.9
Lincomycin	56	57.7	13	13.4	28	28.9
Enrofloxacin	94	96.9	3	3.1	−	−
Gatifloxacin	97	100.0	−	−	−	−

Sensitive—growth retardation of more than 18 mm; Low sensitive—growth retardation of 11–18 mm; Resistant—growth retardation of less than 10 mm; −—negative result.

**Table 5 pathogens-11-00234-t005:** Sensitivity of the isolated microbiota (n = 100) to antibacterial drugs in cats with grade 2 dysbiosis.

Antibacterial Drugs	Antibiotic Susceptibility of Isolated Microbiota					
	Sensitive		Low Sensitive		Resistant	
	Abs	%	Abs	%	Abs	%
Benzylpenicillin	57	57.0	23	23.0	20	20.0
Methicillin	71	71.0	21	21.0	8	8.0
Amoxicillin	77	77.0	17	17.0	6	6.0
Cefazolin	84	84.0	12	12.0	4	4.0
Ceftriaxone	95	95.0	5	5.0	−	−
Cefepim	100	100.0	−	−	−	−
Gentamicin	43	43.0	15	15.0	42	42.0
Lincomycin	54	54.0	17	17.0	29	29.0
Enrofloxacin	94	94.0	5	5.0	1	1.0
Gatifloxacin	100	100.0	−	−	−	−

Sensitive—growth retardation of more than 18 mm; Low sensitive—growth retardation of 11–18 mm; Resistant—growth retardation of less than 10 mm; −—negative result.

**Table 6 pathogens-11-00234-t006:** Sensitivity of the isolated microbiota (n = 107) to antibacterial drugs in cats with grade 3 dysbiosis.

Antibacterial Drugs	Antibiotic Susceptibility of Isolated Microbiota					
	Sensitive		Low Sensitive		Resistant	
	Abs	%	Abs	%	Abs	%
Benzylpenicillin	54	50.4	23	21.6	30	28.0
Methicillin	71	66.4	24	22.4	12	11.2
Amoxicillin	73	68.3	21	19.6	13	12.1
Cefazolin	86	80.4	17	15.9	4	3.7
Ceftriaxone	94	87.9	8	7.4	5	4.7
Cefepim	107	100.0	−	−	−	−
Gentamicin	41	38.3	32	29.9	34	31.8
Lincomycin	49	45.8	32	29.9	26	24.3
Enrofloxacin	89	83.1	10	9.4	8	7.5
Gatifloxacin	107	100.0	−	−	−	−

## Data Availability

Data are contained within the article.

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
