# Peer review of "Fecal Microbiota Analysis in Cats with Intestinal Dysbiosis of Varying Severity"

_pathogens, 2022, doi:10.3390/pathogens11020234_

Round 1

Reviewer 1 Report

Bugrov et al. studied microbiota in cats which various levels of dysbiosis. The topic is interesting and relevant. Additionally, some novel results were obtained on feline microbiota.

However, before I can recommend this paper for publication, several issues have to be addressed. They can be summarized:

1) Authors often use groups, categorizations, criteria, etc., not explained in the M&M section.

2) Although the text grammar is relatively well, the language style is poor with many colloquial expressions. I recommend that a native speaker checks the paper as well.

3) The paper included an experiment on live animals (mice), but the authors did not exhibit clearly its contribution to the study. Therefore, I have some ethical dilemmas.

Major comments:

  1. Page 11, line 324-325

You claim that all cats were fed with the commercial diet (Purina Pro Plan). However, cats were privately owned. A) Was this specific diet your selection criteria for which animals to include? B) I assume that it is a bit risky to claim that all cats were fed just with this food? Do you trust what owners say 100%? Please, elaborate on the topic.

  1. Page 11, line 327

What diagnosis? Dysbiosis? Why was diagnosis important for your study?

  1. Page 13, line 427.

Why did you opt for these antibiotics? Did you study amoxicillin alone or with clavulanate?

  1. Page 2, lines 86-91

What was the frequency of these symptoms in sick cats? Did also healthy cats exhibit any of the symptoms mentioned here?

  1. Page 2, lines 92-99

Unclear. You started talking about Grades, but you did not explain them in the M&M section. You mentioned factors (e.g., alimentary), but you do not define them. You should add an extensive section to M&M and clearly explain your 4 study groups (healthy, Grade 1, 2, and 3). You should state what your criteria for each group are. After M&M is updated, rewrite this section to make it clear.

  1. Table 1

“S. intermedius” – Why did you select to use this name and not pseudintermedius? Explain in M&M (https://www.ncbi.nlm.nih.gov/pmc/articles/PMC3892614/) Due to the techniques you used?

  1. Tables 4, 5, M&M (page 13, lines 430-431)

You mentioned criteria for sensitivities (inhibition zone sizes). Were those related to some standards like EUCAST?

  1. Page 8, lines 204-208.

How were these decreases calculated? Were they simply differences between Tables 4 and 5? In this case, you should replace % with percentage points!

  1. Page 8, Figure 3.

Axis labels are not clear.

  1. Page 9. Figure 4.

Labels for the y-axis are missing. Are [23, 40,…] absolute numbers of isolates? If yes, you are not showing “the ratios,” but only the absolute numbers!

  1. Discussion section.

You mostly comment on your results, but the comparison with other relevant studies is missing!

  1. M&M

You mentioned that you used live mice for bacteria pathogenicity determination; however, I do not see any results from these “experiments” in the paper. What is the correlation? To be this ethically acceptable, one needs to show its contribution to the study.

Minor comments:

  1. Page 1, lines 37-38.

“...,in the conditions of megacities,...” sounds very strangely... You can simply write “in cities” or similar.

  1. Page 11, line 319.

Replace “...” with “.”

  1. Page 12, line 420

What is “mg.”?

  1. Page 6, lines 172, 174, 177

Remove symbol ↓.

  1. Page 7, line 188

In-> by. Remove the arrows.

  1. Page 10, line 288.

Which studies? Cite them.

  1. Page 11, line 330

“sugar centrifugation”? Not a suitable expression! Probably you did fecal centrifugation based on sugar solution?

  1. Page 13, line 439.

Move/copy these p values under relevant plots.

  1. Table 1, Legend. (page 3, line 104).

Mention full bacteria names in the legend since they appear for the first time.

  1. All texts.

Decide if you want to write “grade” or “Grade”.

  1. Page 8, line 225.

What is “opening up”?

Author Response

We would like to thank the respected reviewers for their professionalism and the great work that you put into our manuscript. All comments are objective, reasoned and friendly, thanks to them the article sparkled with new colors. We will try to answer all of the comments.

Major comments:

  1. Page 11, line 324-325

You claim that all cats were fed with the commercial diet (Purina Pro Plan). However, cats were privately owned. A) Was this specific diet your selection criteria for which animals to include? B) I assume that it is a bit risky to claim that all cats were fed just with this food? Do you trust what owners say 100%? Please, elaborate on the topic.

The studies were carried out over a period of three years and the dietary intake served as our criterion for selecting animals for the study. If during the anamnesis collection it turned out that the owners did not comply with our prescriptions, we excluded them from the main groups of the experiment. Appropriate inclusion and exclusion criteria for the experiment are added to the M&M section.

  1. Page 11, line 327

What diagnosis? Dysbiosis? Why was diagnosis important for your study?

Special attention was paid to the diagnosis, for the selection of animals with pathologies of the gastrointestinal tract of various etiologies, which were accompanied by the intestinal dysbiosis of varying severity development. The inclusion criterion was the presence of intestinal dysbiosis.

  1. Page 13, line 427.

Why did you opt for these antibiotics? Did you study amoxicillin alone or with clavulanate?

Antibiotics to determine sensitivity to the isolated microflora were selected based on most frequent use in various pathological processes in cats in veterinary clinics. That's right, we studied amoxicillin without clavulanate.

  1. Page 2, lines 86-91

What was the frequency of these symptoms in sick cats? Did also healthy cats exhibit any of the symptoms mentioned here?

Relevant data on the symptomatology of cats with different severity of intestinal dysbiosis have been added to the manuscript.

  1. Page 2, lines 92-99

Unclear. You started talking about Grades, but you did not explain them in the M&M section. You mentioned factors (e.g., alimentary), but you do not define them. You should add an extensive section to M&M and clearly explain your 4 study groups (healthy, Grade 1, 2, and 3). You should state what your criteria for each group are. After M&M is updated, rewrite this section to make it clear.

Data on the distribution of cats according to the severity of the course of intestinal dysbiosis has been added to the Materials and Methods section.

  1. Table 1

“S. intermedius” – Why did you select to use this name and not pseudintermedius? Explain in M&M (https://www.ncbi.nlm.nih.gov/pmc/articles/PMC3892614/) Due to the techniques you used?

Thank you for your comment. We have added the full name of the types of microorganisms in Table 1. We agree that this makes the material easier to understand. S. intermedius is an anaerobic commensal bacterium that belongs to the group of anginal streptococci. Despite the fact that representatives of this group are commensal microorganisms, some strains may have a wide pathogenic potential. Staphylococcus pseudintermedius is a gram-positive coccal bacterium of the genus Staphylococcus. These are different bacteria. We did not isolate Staphylococcus pseudintermedius.

  1. Tables 4, 5, M&M (page 13, lines 430-431)

You mentioned criteria for sensitivities (inhibition zone sizes). Were those related to some standards like EUCAST?

Thank you for your comment, we added in the MM section that the studies were carried out according to the European Committee on Antimicrobial Susceptibility Testing recommendations (EUCAST; V 8.0, 2020).

  1. Page 8, lines 204-208.

Thanks for your question. We calculated the decrease in the number of sensitive cultures of microorganisms as follows. On the example of benzylpenicillin: the number of sensitive cultures to benzylpenicillin in cats with the grade 1 dysbiosis is 62.9%; the grade 2 - 57.0%, and the grade 3 - 50.4%.

62.9 - 57.0 = 5.9 %

62.9 - 50.4 = 12.5 %

In cats of the second and third experimental groups, a decrease in sensitivity to benzylpenicillin by 5.9% and 12.5% was recorded, when compared with the sensitivity of microorganisms isolated from cats with grade 1 dysbiosis. These differences were discovered by chance, after research, when working with already obtained material. I believe that this interesting trend needs to be further studied using the serial dilution method and more accurate statistical data processing methods. I hope that in further research we will be able to uncover this interesting trend.

  1. Page 8, Figure 3.

Axis labels are not clear.

We tried changing fonts and sizes, but couldn't get the labels to be legible. The drawing file is uploaded separately to the system when submitting to the journal site, and we hope your editors will help to make the captions more clear.

  1. Page 9. Figure 4.

Labels for the y-axis are missing. Are [23, 40,…] absolute numbers of isolates? If yes, you are not showing “the ratios,” but only the absolute numbers!

Thanks for the significant remark, absolutely true, absolute values were used when creating the figure, and in the text, their% ratio was used not to completely duplicate the material of the picture. For greater clarity in the text, we have added absolute numbers to %. Added axis label to figure 4.

  1. Discussion section.

You mostly comment on your results, but the comparison with other relevant studies is missing!

The relevant edits have been made in the Discussion section

  1. M&M

You mentioned that you used live mice for bacteria pathogenicity determination; however, I do not see any results from these “experiments” in the paper. What is the correlation? To be this ethically acceptable, one needs to show its contribution to the study.

Thank you for your comment. Quite right, due to the fact that we did not identify pathogenic cultures of microorganisms, we did not mention these results in the manuscript materials. We added to the “Research Results” section that, according to the results of a biological test on white mice, it was established that all 361 strains of microorganisms that we isolated during the research did not have pathogenic properties and did not cause the death of laboratory animals.

Minor comments:

  1. Page 1, lines 37-38.

“...,in the conditions of megacities,...” sounds very strangely... You can simply write “in cities” or similar.

Thanks for the remark

  1. Page 11, line 319.

Replace “...” with “.”

 Thanks for the remark

  1. Page 12, line 420

What is “mg.”?

This is a translation mistake, not removed from the text.

  1. Page 6, lines 172, 174, 177

Remove symbol ↓.

  Thanks for the remark

  1. Page 7, line 188

In-> by. Remove the arrows.

Thanks for the remark

  1. Page 10, line 288.

Which studies? Cite them.

We meant our research

  1. Page 11, line 330

“sugar centrifugation”? Not a suitable expression! Probably you did fecal centrifugation based on sugar solution?

Of course you're right, it's a mistake

  1. Page 13, line 439.

Move/copy these p values under relevant plots.

Under the graphs there is a corresponding description

Thanks for the remark

  1. Table 1, Legend. (page 3, line 104).

Mention full bacteria names in the legend since they appear for the first time.

 The full names of microbial species have been added to Table 1.

  1. All texts.

Decide if you want to write “grade” or “Grade”.

 Thanks for the remark. We also changed the legend in Figure 4, but these changes are not displayed in review mode

  1. Page 8, line 225.

What is “opening up”?

By this term we would like to say about availability

Once again, we express our extreme gratitude to our reviewers, who took the trouble to study our manuscript in detail, critically comprehend its content, “pass it through themselves”, taking the position of the authors of this work. We are infinitely grateful to you!

Reviewer 2 Report

In this manuscript authors have investigate the quantitative and qualitative fecal microbiota spectrum in cats with intestinal dysbiosis of varying severity. I really appreciate the huge work carried out by the authors. I have few questions that I have to address: 

Major comments:

Did the authors investigate alpha and beta diversity indexes? It is quite relevant evaluating how those parameters are affected in cats having dysbiosis of varying severity versus a control (healthy) condition. 

Minor comments:

Please improve the quality of Fig.1 and Fig.3 and report which stats were used for these graphs.

Author Response

We would like to thank the respected reviewers for their professionalism and the great work that you put into our manuscript. All comments are objective, reasoned and friendly, thanks to them the article sparkled with new colors. We will try to answer all of the comments.

Major comments

Did the authors investigate alpha and beta diversity indexes? It is quite relevant evaluating how those parameters are affected in cats having dysbiosis of varying severity versus a control (healthy) condition.

Thank you very much for the question. We unfortunately did not use alpha and beta diversity indices, thanks for the good idea, in further research we will try to determine the levels of ecosystem diversity in dysbiosis in cats. This will undoubtedly show the extent of this problem.

Minor comments:

Please improve the quality of Fig.1 and Fig.3 and report which stats were used for these graphs.

As far as our competence allows, we have improved the quality of the drawings. The drawings are individually uploaded to the magazine's website, and we hope? Dear editors will help improve their quality. For these graphs, only absolute values and their % ratios were used. For figure 1, in addition, there is a ±95% confidence interval, and for figure 3, additionally, this is a 3D plot that has three categories of values, perhaps due to the presence of three axes of data labels, its quality suffers.

Once again, we express our extreme gratitude to our reviewers, who took the trouble to study our manuscript in detail, critically comprehend its content, “pass it through themselves”, taking the position of the authors of this work. We are infinitely grateful to you!

Round 2

Reviewer 1 Report

Thank you for the changes made in the manuscript.

Reviewer 2 Report

I really appreciated all the changes done by the authors and I have no additional comments.